# Comparing the Effect of Different Management and Rearing Systems on Pigeon Squab Welfare and Performance after the Loss of One or Both Parents

**DOI:** 10.3390/ani9040165

**Published:** 2019-04-14

**Authors:** Azhar F. Abdel Fattah, El-Shimaa M. Roushdy, Hammed A. Tukur, Islam M. Saadeldin, Asmaa T. Y. Kishawy

**Affiliations:** 1Department of Veterinary Public Health, Faculty of Veterinary Medicine, Zagazig University, Zagazig, Ash Sharqia Governorate 44519, Egypt; azharfakhry65@yahoo.com; 2Animal Wealth Development Department, Faculty of Veterinary Medicine, Zagazig University, Zagazig, Ash Sharqia Governorate 44519, Egypt; shimaa_production@yahoo.com; 3Department of Animal Production, College of Food and Agricultural Sciences, King Saud University, Riyadh 11451, Saudi Arabia; tukurhammeda@gmail.com; 4Department of Physiology, Faculty of Veterinary Medicine, Zagazig University, Zagazig 44519, Egypt; 5Department of Nutrition and Clinical Nutrition, Faculty of Veterinary Medic, Zagazig University, Zagazig, Ash Sharqia Governorate 44519, Egypt

**Keywords:** brooding care, pigeon squabs, rearing system, hand-fostering, growth performance, welfare

## Abstract

**Simple Summary:**

Preserving the life and welfare of newly hatched pigeon squabs after the loss of one or both parents will improve pigeon farmers’ income. In the case of losing one parent, the other parent broods the hatched squabs singly. It has however been observed that the growth performance and behavioral development of young squabs provisioned by both parents is improved compared to squabs provisioned by one parent. In case of the loss of both parents, provision by humans (hand-rearing) sustains the lives of squabs. The growth performance and behavioral welfare of squabs provisioned by the hand-rearing method were improved compared to squabs provisioned by foster pigeons.

**Abstract:**

Pigeon squabs completely depend on their parents for care and nourishment. The loss of one or both parents affects squabs’ successful fledging. This study was carried out on young squabs to compare the effect of pigeon parent sex and different fostering methods on squab welfare (behavior and growth performance). Two experiments were carried out. In the first experiment, the squabs were divided into three groups. Group 1 (control) consisted of 10 parent pairs with 20 brooding squabs; group 2 consisted of 10 male parents with 20 brooding squabs; and group 3 consisted of 10 female parents with 20 brooding squabs. In the second experiment, the squabs were also divided into three groups. Group 1 (control) consisted of 10 parent pairs with 20 brooding squabs; group 2 consisted of 20 brooding squabs fostered by 10 foster parent pigeons (either male or female); and group 3 consisted of 20 brooding squabs fostered by the hand-rearing method. A significant improvement in growth performance, behavioral welfare (head waggle, squab note and squab wing shake); increased repetition of these behaviors indicates stress and discomfort), and survival rate was observed to be higher in the group brooded by both parents compared to the group brooded by either a male or a female parent. In addition, the group fostered by hand-rearing showed a significant improvement in growth performance, behavioral welfare, and survival rate compared to the group brooded by foster pigeon parents.

## 1. Introduction

In Egypt, domestic pigeons are reared in towers as domestic livestock. There is a large market for pigeons due to the delicious taste of their meat, which possesses abundant nutrients such as proteins, vitamins, calcium, and iron [1]. Pigeons differ from other domestic birds in their mating and brooding behavior, as young squabs totally depend on their parents for feeding and welfare. Both parents become aggressive during brooding when the eggs have been hatched [2]. Pigeons build two nests for rearing their squabs, and use both nests alternatively for incubating the eggs and for rearing the hatched squabs, controlling a well-timed breeding cycle. Female pigeons reach sexual maturity at about 7 months, eggs are laid 8–12 days after mating, the eggs hatch after 18 days of incubation. The hatched squabs leave the nest within 25–32 days. The breeding cycle in pigeons is about 2 months, when the cycle finished another breeding cycle begins consecutively (https://ovocontrol.com/pigeons/) [3,4]. It is important to understand the behavior of the parents towards their squabs in order to satisfy the squabs’ need for nourishment. This nourishment must satisfy their needs in order for them to show improved body weight, growth rate, and general signs of health [5,6]. Both male and female parents share the responsibility of feeding and taking care of their squabs [7,8]. Newly hatched squabs aged 5–7 days old feed on crop milk secreted by the crop of the parent, which is produced in similar quantities in both parents [9]. Crop milk secretion is stimulated by prolactin and insulin; these hormones affect its weight and content [10]. Crop milk contains high amounts of crude protein and lipids (about 50% and 25–29%, respectively, on a dry matter basis) which improves squabs’ growth performance [11]. The secretion of crop milk continues until about 14–25 days post-hatch [12]. Information on specific dietary requirements of pigeons is limited as most pigeon breeders depend on available grain mixture to feed pigeon flocks [8,13]. A diet of 15–18% crude protein improves pigeon productivity and reproductive performance [14].

In the case of the loss of one or both parents in domestic pigeons, many practices, for example hand feeding, have been established for preserving the squabs’ life, welfare, and performance [15]. In the case of the death of one of the parents, the other parent takes care of the squabs and rears them singly. This has an effect on the squabs’ behavior, welfare, and growth performance [6]. In the case where both parents die, the responsibilities of nurturing the squabs can be transferred to foster parents [16]. Hand feeding of squabs is an attempt to improve livability, welfare, and growth performance [17]. In some pigeon farms, squabs are separated from their parents 3 days after crop feeding to prepare parents for another reproductive cycle. Hand-rearing is the most effective method for rearing squabs and has been proven to be highly successful [18]. The objectives of this work are to compare the effect of pigeon parent sex and different fostering methods on the welfare (behavior and growth performance) of pigeon squabs during brooding in the case of losing one or both parents.

## 2. Materials and Methods

### 2.1. Experimental Design and Husbandry

This study was carried out on a private farm in Menya El-Kameh, El-Sharkia Governorate, Egypt. Egyptian Baladi pigeons were used for this study. The study consisted of two experiments which covered the brooding stage of the squabs (4 weeks). The first experiment was carried out to determine the effect of pigeon parents’ sex on squab behavior and performance. For the first experiment, a total of 40 Egyptian Baladi pigeons consisting of 20 males and 20 females with their 60 brooding squabs were used for this study. The flock was divided into three groups. Group 1 consisted of 20 Egyptian Baladi pigeons with their 20 brooding squabs. Each unit consisted of 1 male pigeon, 1 female pigeon, and 2 brooding squabs. Group 2 consisted of 10 breeding male parents with 20 brooding squabs. Each unit consisted of 1 male parent with 2 brooding squabs. Group 3 consisted of 10 breeding female parents with 20 brooding squabs. Each unit consisted of 1 female parent with 2 brooding squabs.

The second experiment was carried out to determine the effect of different fostering methods on squab behavior and performance. A total of 30 Egyptian adult Baladi pigeons with their 60 brooding squabs were divided into three groups. Group 1 consisted of 10 male and 10 female breeding pigeon parents with their 20 brooding squabs. Each unit consisted of one male and one female pigeon parents with their two brooding squabs (parent-reared). Group 2 consisted of 10 male or female Baladi foster pigeons with 20 brooding squabs. Each unit consisted of either a male or a female adult foster pigeon with two brooding squabs (foster parent-reared). Group 3 consisted of 20 brooding squabs reared by humans (hand-reared). Each unit consisted of two brooding squabs reared by the hand-rearing method (fed and nurtured by humans).

The protocol for the animal experiment was reviewed and approved by the Institutional Animal Care and Use committee at Zagazig University (ANWD–206). For identification purposes, male and female adult pigeons were banded with different colors of fabrics tied (yellow band for male and red band for female) at their wings. Squabs were identified by leg banding. The pigeons were housed in wooden cages (size 55 cm wide, 40 cm deep, and 30 cm high for each unit) placed at different heights on the wall of the building. The researchers ensured that the cages were kept clean, with clean water supplied every day. The sex of squabs was determined by pelvic bone test. The squabs were held in an upright position, and then the researcher(s) ran the index finger between the legs to the vent area. If the two bones (which are close together) at the cloacae are hard, it is a male pigeon. If there are curved, spongy bones which are rounded at the tip and the finger fits between the ends, it is a female. Nesting materials were provided in form of hay or soft dried grasses about 4–6 inches in the length. Each nest contained two white oval eggs. The first eggs were laid about 6–10 days after the adult pigeon pairs were mated. The second eggs were laid about 26–48 h after the first set of eggs were laid. Both parents shared the responsibility of incubation and brooding of hatched squabs.

### 2.2. Feeding and Diet Formulation

Both parents fed the squabs from pigeon milk secreted by the crop during first few days of life. Thereafter, pigeon parents added previously digested feed to the diet of their squabs at 6–10 days. Parent pigeons were fed twice daily. Pigeon rations were formulated according to [19]. The ration ingredient was grounded, mixed and pelleted to suitable sizes for adult pigeons (Table 1).

#### 2.2.1. Foster Parent Pigeon Rearing

The foster parent was chosen of the same Egyptian Baladi breed, and was in the breeding cycle, brooding its own hatched squabs. We introduced the orphan hatched squabs to the foster parent pigeon nest to be brooded by this foster parent. The foster parent secreted crop milk, but introducing hatched orphan squabs to foster parent nest increased the aggressive behavior of the parent toward those squabs, at times with neglect in feeding.

#### 2.2.2. Hand-Feeding Protocol

For hand feeding of orphan squabs, the same ration was used, but in mashed form. The feed was moisturized with warm water until it reached the required consistency. The moisturized feed was sized into a pellet size of 4 mm. Utmost care was taken to feed the squabs with a rubber tube which ensured that the doughy feed was introduced into the crop of the squabs and not to the lungs. To achieve this, a rubber tube of diameter 4 mm and length 3–3.5 inches was used. To feed the squabs, each squab was gently held, with no pressure applied, over the crop area. The neck of squabs straightened vertically, with the beak open. This eased the introduction of the rubber tube. The rubber tube, previously lubricated with Vaseline, was gently passed into the squabs’ esophagus until it reached the crop. The tube was gently felt by hand from outside to ensure it had reached the crop. Warm liquid feed was fed to the squabs. Care was taken to ensure that feed was neither hot nor cold, just slightly warm. This ensured that the squabs were not irritated by the feed, and hence did not regurgitate it. Squabs adapted quickly to opening the mouth for the syringe which differs from the natural ways of eating. When squabs grow older, they place their beaks in their parents’ beaks to suck out regurgitated food from the mouth. Consequently, as the hand-reared squabs grew, the tube was placed in their beaks instead of the crop. Hence, the squabs sucked the feed from the tube itself. Hand feeding was applied every 2 h daily from 06:00 to 18:00 h during the experiment (8 times daily), filling the crop during the daylight hours and leaving empty at night. Overfeeding the squabs may lead to regurgitation, which may in turn lead to aspiration and death.

### 2.3. Observation of Squabs’ Behavior

The behavioral observations were performed using a focal sample technique [21], an observation sheet for recording behaviors based on an ethogram (see below), a stop watch, and digital camera according to [22]. All the experimental groups were observed for a total of 2 h per day. Each individual squab was observed for 3 min daily and the observation was continuous to record event behaviors, detailed in the following ethogram.

(1)Clutching: This is a fear or stress behavior, as squabs cling to their parents’ toes of with their feet if they are disturbed in order to be close to them.(2)Head waggle (shaking): Squabs shake or waggle the head as if the muscles of neck are unable to carry the head up; they exhibit this behavior while they are seeking for feed or new positions.(3)Squab note: Hungry squabs give a prolonged shrill with an ascending whistle of low intensity for a duration of 1–3 s, or longer in older squabs. The sound is often repeated several times and is accompanied by wing shake and bill searching.(4)Squab wing shake: Squabs shake their wings about 2–4 times per second. This is done repeatedly to show that the squab needs to be fed. This movement increases to be more obvious during the first days of life during brooding.(5)Bill searching: Older squabs direct their beak forward and toward both male and female pigeon in rapid ‘wiggling’ movements.(6)Nest defecation: As the squab increases in age (pin feathers developing) they defecate over the edge of the nest. When pin feathers are fully developed they move away from the nest to defecate.(7)Bill snapping: Squabs click or snap their bills one or more times when closely accosted by an unfamiliar object or movement.(8)Hissing and puffing: Squabs may fluff the feather especially the breast and “hiss” when accosted by an unfamiliar object or movement. They also flap their wings as a sign of flight stance to appear bigger in size and stand in biting position as a defense mechanism.(9)Squeaker notes: Squabs at 4 weeks of age or younger produce squeaky notes which adult pigeons interpret to be notes of alarm to inform their parents.

### 2.4. Growth Performance

All newborn squabs were weighed early in the morning (07:00 h) before feeding. They were weighed every day until the end of brooding stage (4 weeks). The body weight gain (BWG), average feed intake (AFI) was calculated in the case of parent-fed squabs by weighing the squabs before feed supplementation and reweighing after being fed by the parent, with subtraction of the two weights to get the amount of feed intake for the squab, and then calculation of the total feed intake per day for each squab. In case of hand feeding, the amount of feed for squab was weighed. The researchers ensured that the squabs were not fed for 2 h before the commencement of weighing. Feed conversion was calculated as feed intake (g)/total gain (g). Live body weight (LBW) and BWG were evaluated based on individual bird data, whereas AFI and feed conversion ratio (FCR) were assessed based on the replicate unit. The body weight gain (BWG) was calculated by subtracting final weight from initial weight of chicks. The FCR was calculated for each squab as feed intake divided by body weight gain and then statistically calculated for each group as described by [23]. The relative growth rate (RGR) was calculated using the following equation: (½ (Final weight – initial weight)/(Final weight+initial weight)×100), as described by [24]. Protein efficiency ratio (PER) was determined according to [25] as the number of grams of weight gain produced per unit of weight of dietary protein consumed. Survival rate was calculated directly according to [26] as the percentage of surviving squabs with respect to the total number of squabs in each group. Each two squabs were considered replicate with 100% survival if both were still alive, 50% of the unit if one squab died during the experiment, and 0% if both squabs of the replicate had died and each group contained 10 replicates.

### 2.5. Statistical Analysis

All statistical analyses were performed using SPSS V.16 software (SPSS Inc., Chicago, IL, USA). Data were analyzed using a one-way ANOVA, using model Y iK= µ+Li+eiK, where Y ik = observation, µ = over all means, and Li = effect of groups as G1 (group 1) (j = 1, 2, and 3); in the first experiment, 1 = parents-reared, 2 = male parent-reared, and 3 = female parent-reared. For the second experiment, 1 = parents-reared, 2 = foster pigeon-reared, and 3 = human-reared. eik = random error, after normality was verified using the Kolmogorov–Smirnov test. Duncan multiple range tests were used to determine significant differences between mean values [27]. Variability in the data was expressed as the standard error of mean (SEM), and the alpha level for significance was set at 0.05. The effect of replicate within the same group was analyzed and indicated no significant difference among replicates. Data for each squab were calculated individually, with total number of squabs 20 within the group and a total 60 for all three groups in each experiment.

## 3. Results

### 3.1. Effect of Pigeon Parent’s Sex on Squab’s Behavior During Brooding Period

In case of losing one parent, pigeon parent sex affects squab behavior during the brooding period Table 2. The results show that there was a significant increase at total degree of freedom of 59 (between groups 2 and within groups 57) in clutching (26.09, 20.11, and 11.07), head waggle (34.11, 24.12, and 10.10), and hissing and puffing (28.13, 20.18, and 7.18) in the group brooded by males, the group brooded by females, and the group brooded by both parents, respectively. Squab note, squab wing shake, bill searching, bill snapping, nest defecation, and squeaker note behaviors were not significantly different between groups brooded by either males or females; at the same time those parameters were significantly higher in groups brooded by either males or females compared to the group brooded by both parents.

### 3.2. Effect of Different Fostering Methods on Squab’s Behavior during Brooding Period

The effect of different fostering methods on squab behavior during the brooding period is presented in Table 3. With a total degree of freedom (DF) of 59 (between groups 2 and within groups 57) clutching, head waggle, squab note, squab wing, bill searching, bill snipping, hissing and puffing and squeaker note behaviors showed significantly higher values in the foster pigeons-reared group, hand-reared group, and then the parents-reared group. Meanwhile, nest defecation was recorded as significantly higher in the parents-reared group and hand-reared group as compared to the foster pigeons-reared group.

### 3.3. Effect of Pigeon Parent’s Sex on Growth Performance of Squabs during the Brooding Period

The effect of pigeon parents’ sex on growth performance of squabs during the brooding period in the case of losing one parent was studied. The results (Table 4) showed no significant differences at total degree of freedom of 59 (between groups 2 and within groups 57) in final body weight, total weight gain, total feed intake (FI), feed conversion ratio (FCR), and relative growth rate (RGR) in groups brooded by one parent (either male or female); at the same time these parameters were significantly lower in groups brooded by one parent (either male or female) when compared to the groups brooded by both parents. On the other hand, the protein efficiency ratio (PER) was significantly higher in groups brooded by one parent (either male or female) compared to the group brooded by both parents. Regarding the survival rate for groups brooded by one parent (either male or female), there was a significantly lower survival rate at 95% (degree of freedom 29; each two squabs were considered one replicate, with 10 replicates for each group) compared to that for the group brooded by both parents (100%).

### 3.4. Effect of Different Fostering Methods on Growth Performance of Squabs during Brooding Period

The effect of different fostering methods on squab growth during the brooding period is indicated in Table 5. At a total degree of freedom of 59 (between groups 2 and within groups 57), the final body weight, total gain, FI, FCR, and RGR were significantly increased in the parents-brooded group, hand-reared group, and foster pigeon-reared group. In the same vein, PER significantly increased in the foster pigeon-reared group, the hand-reared group, and then the parents-brooded group (2.67, 1.86, and 1.63, respectively). The hand-reared group recorded a significantly higher survival rate of 85% (degree of freedom 29; each two squabs were considered one replicate, with 10 replicates for each group) compared to the foster pigeon-reared group (65%) but a significantly lower survival rate as compared to the two parent-reared group (100%).

## 4. Discussion

Pigeons differ from other domestic birds in their mating and brooding behavior. Young squabs totally depend on their parents for feeding and brooding. Male and female pigeons share the responsibility for taking care of their squabs [7,8]. In the case of the loss of one or both parents, the squabs’ lives may be affected, or at the very least their behavior and performance. Our results showed significant increase in clutching, head waggle and hissing, and puffing, in the group brooded by males, the group brooded by females, and then the group brooded by both parents. Squab note, squab wing shake, bill searching, bill snapping, nest defecation, and squeaker note behavior was not significantly different between the groups brooded by either males or females; at the same time these parameters were significant higher in groups brooded by either males or females compared to the group brooded by both parents. These behaviors appear to be more frequent in groups brooded by one parent, and this may be due to lower frequencies of feeding of these squabs. This lowered feeding rate may increase stress on squabs, leading to increased frequencies of these stress behaviors when compared to squabs reared by two parents.

Squab behavioral stimuli are related to the parent–squab interaction [6], which affects squab behavior [28]. Both parents partition the time allotted to parental care activities; with greater frequencies of feeding the squab’s stress behaviors are not stimulated, and better welfare is achieved [29]. Squab behavior stimulates the secretion of prolactin, which plays an important role in crop milk secretion for feeding the hatched squabs [10]. The results are reinforced by the authors of [22], who reported that hanging behavior was increased in stressed squabs and adult pigeons, indicating poor welfare. The results of this study indicated that there was a significant increase in nest defecation in the group brooded by both pigeon parents compared to the group brooded by one pigeon parent. This was related to higher frequency of feeding by both parents than one parent.

In case of the loss of both parents, the fostering method (foster pigeon parents or hand-rearing by humans) also affects squab behavior during the brooding period. The results of our study showed that clutching, head waggle, squab note, squab wing, bill searching, bill snipping, hissing and puffing and squeaker notes behavior was significantly higher in the foster pigeons-reared group, hand-reared group, and then the parents-reared group. Meanwhile nest defecation was recorded as being significantly higher in the parents-reared group and hand-reared group compared to the foster pigeons-reared group. This may be due to foster pigeon’s refusal of squabs or reduction of the frequency of feed regurgitation, which leads to an increased hunger level. In addition, there was a significant decrease in nest defecation in the group brooded by foster pigeons compared to other groups. This may be due to the low level of feeding activity from foster pigeon towards squabs. In contrast, Hansen [30] reported improved crop development and feeding of squabs when they were introduced to foster parent pigeons after their own eggs hatched.

Hand-reared squabs showed lower clutching, head waggle, squab note, squab wing, bill searching, bill snapping, hissing, puffing, and squeaker notes behaviors compared to the group brooded by foster pigeon parents. Our result was enforced by finding of Yang and Vohra [31] who reported that hand-rearing of squabs can improves performance and meat their feed requirement which decrease squabs hanger feeling consequently decrease stress behavior and improve welfare.

Growth performance parameters showed no significant differences in final body weight, total weight gain, total feed intake (FI), feed conversion ratio (FCR), and relative growth rate (RGR) in groups brooded by one parent (either male or female); at the same time, these parameters were significantly lower in groups brooded by one parent (either male or female) when compared to the group brooded by both parents. On the other hand, the protein efficiency ratio (PER) was significantly higher in groups brooded by one parent (either male or female) compared to the group brooded by both parents. Research that explains the reasons for the low conversion rate of feed in young pigeon squabs brooded by both parents is limited. The presence of both parents could be associated with an increase of squab feeding frequency; thus, the body weight and total weight gain increase while PER decrease because of high feed intake. The survival rate was higher in group brooded by both parents than other groups; this may due to the intensive care exhibited by parents to achieve high welfare to their squabs.

In case of the loss of both parents, the fostering method affects the growth performance of squabs. Final body weight, total gain, FI, FCR, and RGR were significantly increased in parents-brooded group, hand-reared group, and foster pigeon-reared group. In the same vein, PER significant increases in the foster pigeon reared group, the hand-reared group, and then the group brooded by both parents. The hand-reared group recorded a significantly higher survival rate than the foster pigeon-reared group, and a significantly lower survival rate than the both parents-reared group.

This may be due to foster pigeons neglecting to feed strange squabs, causing retardation in all growth parameters. This corresponds with the data recorded by Harrington et al. [17] and Yang and Vohra [31] who supported the finding that hand-feeding of pigeon squabs improves survival, growth performance, and welfare. This is inconsistent with another study [30] which reported that brooding squabs by foster pigeons after hatching their own eggs improved squab performance and growth. The survival rate was improved in the hand-reared group as compared to the foster pigeon-reared group, but lower than in the both parent-reared group. Thus, hand-rearing may achieve more adequate feeding of squabs as compared to rearing by foster pigeons, but does not reach the level achieved by their parents. Many measures form part of animal welfare, with poor welfare indicated by retardation of growth, behavioral anomalies, high mortalities, immunosuppression, and diseases, etc. [32]. This study measured the welfare of squabs stressed by losing one or both parents through the application of different rearing protocols, and measured the improvement in squabs’ welfare by measuring frequencies of stress behaviors, growth performance, and survival rate. Our recommendation from this study is that pigeon breeders should take care of young squabs since one pigeon parent cannot adequately care for young squabs. This is important to ensure adequate growth performance and welfare. Human interference by the hand-rearing method solves the problem of decreased growth rate, prevents mortality, reduces economic loss, and increases outcome. In the case of the loss of both parents, it is recommended to foster young squabs by hand-rearing instead of the use of foster parents, who may lack the motivation to adopt an unfamiliar squab. In the same light, in the case of the loss of both parents for any reason, it is recommended to foster the squabs by hand-rearing due to the difficulty of fostering by foster parents with ill-developed motivation to adopt strange squabs.

## 5. Conclusions

In the case of the loss of one or both pigeon parents, the foster method (foster pigeon parent or hand-rearing by humans) affects the growth performance and behavior of squabs. A foster method is however useful since it sustains the lives of young squabs. Hand-feeding by humans is shown to be more effective compared to the use of foster pigeon parents. Hand-reared squabs showed better signs of health than foster-parented squabs.

## Figures and Tables

**Table 1 animals-09-00165-t001:** Ingredients and chemical composition of the basal diet (as-fed basis).

Ingredients	%
Yellow corn	25.00
Soybean meal, 48%	8.00
Sorghum grain	13.7
Wheat grain	23.45
Chicken pea	10
Broad beans	8
Wheat bran	8
Calcium carbonate	1.00
Dicalciumphosphate	2.00
Premix *	0.30
DL-Methionine, 98%	0.25
Lysine, HCl, 78%	0.30
Calculated chemical composition †	
† ME, kcal/kg	2760.73
CP, %	15.98
EE, %	2.12
CF, %	4.26
† Ca, %	1.001
† Available P, %	0.56
Lysine, %	0.92
Methionine, %	0.43

* Supplied per kg of diet. Vitamin A, 12,000 IU; vitamin D3, 2200 IU; vitamin E, 26 IU; vitamin K3, 6.25 mg; vitamin B1, 3.75 mg; vitamin B2, 6.6 mg; vitamin B6, 1.5 g; pantothenic acid, 18.8 mg; vitamin B12, 0.31 mg; niacin, 30 mg; folic acid, 1.25 mg; biotin, 0.6 mg; Fe, 50 mg; Mn, 60 mg; Cu, 6 mg; I, 1 mg; Co, 1 mg; Se, 0.20 mg; Zn, 50 mg; choline chloride, 500 mg. † Calculated according to [20]. ME, metabolic energy; CP, crude protein; EE, ether extract; CF, crude fiber; Ca, calcium; P, phosphorus.

**Table 2 animals-09-00165-t002:** Effect of pigeon parent sex on squab behavior during the brooding period.

Parameters (Frequencies All over the Experiment)	Groups			SEM	*F*-Value	*p*-Value
	Brooding by both Parents	Brooding by Males Only	Brooding by Females Only			
Clutching	11.07 ^c^	26.09 ^a^	20.11 ^b^	0.10	2582.02	0.00
Head waggle	10.10 ^c^	34.11 ^a^	24.12 ^b^	0.10	4963.14	0.00
Squab note	8.06 ^b^	21.11 ^a^	21.11 ^a^	0.13	857.59	0.00
Squab wing shake	9.13 ^b^	14.11 ^a^	14.11 ^a^	0.14	183.52	0.00
Bill searching	8.27 ^b^	19.13 ^a^	19.17 ^a^	0.07	4080.35	0.00
Nest defecation	10.19 ^a^	7.08 ^b^	7.17 ^b^	0.10	157.45	0.015
Bill snapping	10.11 ^b^	25.11 ^a^	25.16 ^a^	0.07	480.16	0.003
Hissing and puffing	7.18 ^c^	28.13 ^a^	20.18 ^b^	0.12	4010.08	0.00
Squeaker notes	9.19 ^b^	23.09 ^a^	23.11 ^a^	0.12	2080.45	0.00

^a–c^ Means bearing different superscript letters within the same row are significantly different (*p* <0.05) SEM: Standard error of the means.

**Table 3 animals-09-00165-t003:** Effect of different fostering methods on squab’s behavior during the brooding period.

Parameters (Frequencies All over the Experiment)	Groups			SEM	*F*-Value	*p*-Value
	Parents-Reared	Foster Pigeons-Reared	Human-Reared			
Clutching	10.2 ^c^	25.33 ^a^	15.13 ^b^	0.21	650.16	0.00
Head waggle	9.13 ^c^	33.10 ^a^	20.10 ^b^	0.11	4100.1	0.00
Squab note	9.10 ^c^	35.16 ^a^	22.19 ^b^	0.11	6830.24	0.00
Squab wing shake	8.12 ^c^	23.15 ^a^	22.17 ^b^	0.12	2790.4	0.013
Bill searching	11.13 ^c^	34.14 ^a^	33.10 ^b^	0.09	391.59	0.002
Nest defecation	8.11 ^a^	5.00 ^b^	8.00 ^a^	0.12	96.03	0.00
Bill snapping	9.09 ^c^	31.15 ^a^	27.13 ^b^	0.13	2330.14	0.00
Hissing and puffing	6.08 ^c^	28.11 ^a^	27.13 ^b^	0.08	4160.26	0.00
Squeaker notes	8.08 ^c^	30.0 ^a^	27.11 ^b^	0.10	7660.07	0.00

^a–c^ Means bearing different superscript letters within the same row are significantly different (*p* <0.05) SEM: Standard error of the means.

**Table 4 animals-09-00165-t004:** Effect of pigeon parent’s sex on growth performance of squabs during the brooding period.

Parameters	Groups			SEM	*F*-Value	*p*-Value
	Brooding by both Parents	Brooding by Male Only	Brooding by Female Only			
Initial body weight (g)	49.65	48.55	47.05	0.58	1.70	0.191
Final body weight (g)	467.50 ^a^	290.90 ^b^	287.25 ^b^	10.98	5878.55	0.00
Body weight gain (g)	417.85 ^a^	242.35 ^b^	240.20 ^b^	10.86	7743.08	0.00
Total feed intake (g)	1394.20 ^a^	710.25 ^b^	680.45 ^b^	43.99	562.85	0.00
Feed conversion ratio	3.33 ^a^	2.93 ^b^	2.83 ^b^	0.04	23.16	0.00
Relative growth rate	161.68 ^a^	142.8 ^b^	143.74 ^b^	1.22	173.62	0.00
Protein efficiency ratio	1.88 ^b^	2.16 ^a^	2.22 ^a^	0.03	17.94	0.00
Survival rate %	100.00	95.00	95.00	2.32	0.50	0.612

^a–c^ Means bearing different superscript letters within the same row are significantly different (*p* <0.05). SEM: Standard error of the means.

**Table 5 animals-09-00165-t005:** Effect of different fostering methods on growth performance of squabs during brooding period.

Parameters	Groups			SEM	*F*-Value	*p*-Value
	Parents-Reared	Foster Pigeons-Reared	Human-Reared			
Initial body weight (g)	49.05	49.11	49.31	0.06	1.67	0.197
Final body weight (g)	469.55 ^a^	277.35 ^c^	302.70 ^b^	11.12	7047.83	0.00
Body weight gain (g)	420.51 ^a^	228.24 ^c^	253.39 ^b^	11.13	6715.51	0.00
Total feed intake (g)	1617.30 ^a^	536.20 ^c^	852.85 ^b^	59.15	11838.63	0.00
Feed conversion ratio	3.85 ^a^	2.35 ^c^	3.37 ^b^	0.08	1341.16	0.00
Relative growth rate	162.17 ^a^	139.82 ^c^	143.94 ^b^	1.27	2628.48	0.00
Protein efficiency ratio	1.63 ^c^	2.67 ^a^	1.86 ^b^	0.06	1132.34	0.00
Survival rate %	100.00 ^a^	65.00 ^b^	85.00 ^ab^	5.54	4.06	0.029

^a–c^ Means bearing different superscript letters within the same row are significantly different (*p* <0.05). SEM: Standard error of the means.

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
