# Peer review of "Comparing the Effect of Different Management and Rearing Systems on Pigeon Squab Welfare and Performance after the Loss of One or Both Parents"

_animals, 2019, doi:10.3390/ani9040165_

Round 1

Reviewer 1 Report

The study aimed to evaluate the effects of management and feeding systems on pigeon squabs’ welfare and growth performance. There are some disadvantages that need to be corrected, before it is suitable for publication.

1.      Lines 51: Change “[1].[7]” to “[1]”.

2.      Why you using the same ration for the adult pigeons and the hand feeding of orphan squabs? The nutrient requirement for the adult pigeon and squabs is different. You should make the ration according to the nutrients value of crop milk.

3.      Line 134: Please add space between letters and units. For example, change “4mm” to “4 mm”.

4.      Line 183: For the FCR, I don’t think the formula is correct. You cannot use the feed intake of parents divided by body weight gain of young squabs as the FCR.

5.      Line 194: Have the survival rates of squabs been determined? It is an important index. Please add the information.

6.      In Table 2: Please check the value of “Hissing and puffing” is right.

7.      In Table 2 and 3: Please use the superscript for the letters. Making it consistent with Table 4 and 5.

8.      Lines 217 – 219: How to control the feed intake of squabs in experiment 2? Please clarify.

Author Response

We appreciate the time, efforts, and overall constructive criticism raised by the reviewer. All the suggestions have been addressed, and corrections made where necessary.

Query:

1.       Lines 51: Change “[1].[7]” to “[1]”.

Response:  corrected to (1).

2.       Why you using the same ration for the adult pigeons and the hand feeding of orphan squabs? The nutrient requirement for adult pigeon and squabs is different. You should make the ration according to the nutrients value of crop milk.

Response:   We used the same ration for adult pigeon and hand feeding orphan squabs because the aim, in this case, is to keep reasonable performance and life of these squabs as the formation of ration containing about 50% cp and 25% lipids on dry matter base was not economic and not practical for the farm.

3.       Line 134: Please add space between letters and units. For example, change “4mm” to “4 mm”.

Response:   corrected to (4 mm).

4.       Line 183: For the FCR, I don’t think the formula is correct. You cannot use the feed intake of parents divided by body weight gain of young squabs as the FCR.

Response:   FCR formula is feed intake g/ total weight gain g and I clarify how I calculated the feed intake of the parent fed squab by weighting the squab before supplementation of feed and after being fed by the parent, then subtracting the two weights to get the amount of feed intake.

5.       Line 194: Have the survival rates of squabs been determined? It is an important index. Please add the information.

Response:  added to performance tables 4, 5.

6.       In Table 2: Please check the value of “Hissing and puffing” is right.

Response:   corrected.

7.       In Table 2 and 3: Please use the superscript for the letters. Making it consistent with Table 4 and 5.

Response:   corrected.

8.       Lines 217 – 219: How to control the feed intake of squabs in experiment 2? Please clarify

Response:   clarified in material and method (growth performance calculation). 

Reviewer 2 Report

There is merit to this paper as it provides some useful measures of bird welfare that are not always readily available in the literature. The paper needs work to improve clarity and ensure that results fully comprehensible.

Perhaps do not mention welfare until the discussion. You do not measure welfare per se. You collect data on behaviour and growth rates. You give no explanation of how these inform positive welfare states and how to identify them. Maybe explain what these behaviours / growth parameters mean for positive welfare in your discussion?

Simply summary

Is it really fostering when a pigeon squab is hand-reared? 

Squabs are not brooded by human carers, they are provisioned. So you are looking at differences in provisioning between two parent, one parent, foster parent and human carers.

Abstract

Too based on methods. This needs to be an overall summary of the whole experiment. Please cut down the description of groups and how data were collected and add in more information results and their discussion / evaluation.

 Line 30: depend on their parents for care and nourishment, not sustainability 

Line 31: changes of successful fledging, not welfare

Lie 43: how have you measured "behavioural welfare"?

Introduction

Very repetitive and poorly explained. Can you cut out some of the superfluous information and focus on behavioural measures of welfare in pigeons and references that support why you are conducting this experiment? There is a lot of repetitive information on crop milk. And who produces it and why it is produced. Slim this down and give more focus to research papers that support your aims and hypotheses. 

Line 49: Not really needed. 

Line 50 Needs a reference.

Best to not start a sentence with "They..." it is vague. Be specific to the item, animal, author or research you are referring.

Line 52: are pigeons really poultry? Maybe say they different from other domestic birds in that...

Line 54: what are these natural welfare needs?

Line 55: building of two nests and the function of this needs a reference.

Line 56: what is a well-timed breeding cycle?

Line 57: surely in the wild if a squab lost both parents it would die? Only in domestic pigeons would someone step in to rear youngsters who had no parents. Make it clear that this is behaviour in a domestic sense.

Line 63/64: repetition about crop milk production.

Line 73: "good welfare" instead of just welfare.

Line 74: not breeds them singly, provisions/rears them singly. 

Line 75: squabs' behaviour.

Line 76: you introduce parents being removed from the chick but the reason is not given until later on. Move this section on why farmers would do this for earlier on because it justifies why you are running this experiment.

Line 82: is growth performance akin to welfare? Think about physical and behavioural measures of welfare and what is measureable.

Materials and methods

Generally well written and the experimental design of how birds are included in different groups is easy to follow. Perhaps restructure and give subheadings. Explain and describe the sample population and its husbandry, housing (including sizes of cages) and nutrition first. Then explain the experimental set-up and what was tested on what (i.e. behavioural measures of welfare and growth /development parameters). Have a section with a subheading about the hand-rearing protocol. Then finished with a paragraph on statistical analysis.

Were the foster pigeons that were used the same breed? And where they also in a reproductive phase? I.e. did they have eggs and nests? How did you get these foster pigeons to produce crop milk?

I'm still unsure whether the human carers are really fostering? Perhaps you describe your groups as i) parent reared, ii) foster parent reared (pigeon), iii) hand-reared (human).

Line 114: nesting material was provided in the form of...

Line 121 and 122: please check your use of referencing as this hard to follow. Is this the style of the article? It would be easier to refer to a specific author so that you construct a proper sentence I.e. Pigeon parents were fed a standard pigeon ration as per Smith et al. [reference number].

Is table 1 needed? Surely this can be looked up from the reference about diets? Did you do any nutritional analysis that would warrant a breakdown of ingredients in this manner?

Line 139: Vaseline is petroleum jelly. Is this safe to use on something an animal is ingesting?

Line 145: how do pigeon squabs come of age?

Line 149: please include your ethgram of behavioural defintions somewhere near here.

Line 151 and 152: I don't understand... each pigeon was watched for two hours? What's the three minutes for? If you do a focal sample, is this instantaneous or continuous recording? How often were behaviours noted? And did you focus on state or event behaviours? This needs more explanation. 

Line 153: according to whom? Is this a subheading? Please work on your layout.

Line 154: this behavioural description does not make sense. I would not be able to observe this behaviour myself based on this definition and description.

Line 158: as above. Think about the name of this behaviour too.

Line 166-167: Re-write "As the squab increases with age (pin feathers developing) they defecate over the edge of the nest. When pin feathers are fully developed they move away from the nest to defecate". 

Line 169: as above. I don't understand this behaviour.

Line 172: How do you know this is why they do this?

Line 174: again, how do you know this is why they do this?

Line 185 and 186: strange referencing again / hard to read sentence structure.

Line 190: Only an ANOVA? No modelling was attempted to pick out the influences of different variables? Seems quite limited statistical analysis for an elaborate and complex experimental design.

Line 191: What do you mean? ref: no sig. diff.

Duncan multiple range tests WERE used...

Results

You need to present statistical output correctly in the text (i.e. the test statistic, degrees of freedom, r2 value- if appropriate, and the P value). You do not provide the reader with any sense of the validity of your testing by just including P values.

Think about whether or not you have really measured welfare... You measure behaviour and you collect information on growth. But you give no link to welfare. Are there any welfare-friendly behaviours that you would look out for that tell you pigeon squabs are experiencing positive welfare states?

Please re-write the whole results section so that inferential analysis is presented around descriptive analysis for each hypothesis tested. 

Table 2: You are presenting multiple P values together for comparison and therefore you should check for Type I error, I suggest a Benjamini-Hochburg to correct your alpha level. See: Benjamini, Y. and Hochberg, Y. (1995) Controlling the False Discovery Rate: A practical and powerful approach to multiple testing. Journal of the Royal Statistical Society B 57 (1): 289–300. 

You should be able to find an automatic calculator for this online.

Also, for table 2, what are these data being tested? Time? Frequency? Counts of behaviour? 

Table 3: I don't understand how these are growth indicators? They are behavioural categories again? Please clarify or change what data you include.

Table 4: This is understandable. I see what is being analysed here. Again, work on statistical output.

Table 5: humans do not brood.

Discussion

Please restructure. Start off with an explanation of your findings. Then tell the reader what they mean. And then add extensions. And finally your conclusion.

What are your key results and why? How do they compare to the background literature? How do they add to it?

At the moment, your discussion is hard to follow and does not "sell" the results of your own experiment to the reader.

Some of the discussion is very descriptive and simply explains pigeon natural history and physiology. You are not evaluating findings against published literature. Try to base your discussion on your results and synthesise new meaning from these results via comparison or contrast with published work. Or if your results are novel, and there is no comparison in the literature, evaluate why this is. Don't tell me about the physiology of how pigeons make crop milk, for example.

I am confused as to how foster pigeons were selected? Where they in breeding condition? In avicultural circles, pigeons make excellent foster parents so long as they are already nesting an they can be given eggs to hatch. Is the poor performance of foster parents because they were given already hatched chicks? Please explain this better. 

Line 267: "Squab behaviour stimulates..."

Line 299: "This corresponds with data recorded by *Give author name*..."

Line 300: improves survival

Line 305: showed to be, not proved.

Are the recommendations better embedded as a final paragraph of the discussion before the conclusion?

Line 314: hatched, not birthed.

As I said previously, describe the repro. state of the foster pigeons. If they are in the right physiological state they will make excellent foster carers. 

Author Response

We thank the reviewer for excellent revewing and very useful sugggestions to improve the quality of the mansucript.

Simply summary

Is it really fostering when a pigeon squab is hand-reared? 

Squabs are not brooded by human carers, they are provisioned. So you are looking at differences in provisioning between two parent, one parent, foster parent and human carers.

Response:   corrected.

Abstract

Too based on methods. This needs to be an overall summary of the whole experiment. Please cut down the description of groups and how data were collected and add in more information results and their discussion / evaluation.

Response:  more summarized.

 Line 30: depend on their parents for care and nourishment, not sustainability 

Response:  corrected.

Line 31: changes of successful fledging, not welfare

Response:   corrected.

Lie 43: how have you measured "behavioral welfare"?

Response:   behavioral welfare measured by all behavioral parameters measured, as Head waggle, Squab Note, Squab wing shake, increase the repetition of these behaviors indicate stress and uncomfortable squab.

Introduction

Very repetitive and poorly explained. Can you cut out some of the superfluous information and focus on behavioural measures of welfare in pigeons and references that support why you are conducting this experiment? There is a lot of repetitive information on crop milk. And who produces it and why it is produced. Slim this down and give more focus to research papers that support your aims and hypotheses. 

Response: generally the repetition removed and summarized.

Line 49: Not really needed. 

Response: removed

Line 50 Needs a reference.

Response: reference no. (1).

Best to not start a sentence with "They..." it is vague. Be specific to the item, animal, author or research you are referring.

Response: corrected.

Line 52: are pigeons really poultry? Maybe say they different from other domestic birds in that..

Response: corrected.

Line 54: what are these natural welfare needs?

Response: corrected.

Line 55: building of two nests and the function of this needs a reference.

Response: reference no. (3, 4).

Line 56: what is a well-timed breeding cycle?

Response: as female pigeon reach sexual maturity at about 7 months, the eggs laid 8-12 days after mating, the eggs hatch after 18-day of incubation, finally the hatched squab leave the nest within 25-32 day. The breeding cycle in pigeon is about 2 months; when the cycle finished another breeding cycle begin consecutively.

Line 57: surely in the wild if a squab lost both parents it would die? Only in domestic pigeons would someone step in to rear youngsters who had no parents. Make it clear that this is behaviour in a domestic sense.

Response: corrected.

Line 63/64: repetition about crop milk production.

Response: the repetition removed.

Line 73: "good welfare" instead of just welfare.

Response: corrected.

Line 74: not breeds them singly, provisions/rears them singly. 

Response: corrected.

Line 75: squabs' behaviour.

Response: corrected.

Line 76: you introduce parents being removed from the chick but the reason is not given until later on. Move this section on why farmers would do this for earlier on because it justifies why you are running this experiment.

Response: removed (some farmers remove squabs early from their parent to introduce the parent in another breeding cycle more early and feed squabs by hand rearing).

Line 82: is growth performance akin to welfare? Think about physical and behavioural measures of welfare and what is measureable.

Response: the researchers measured the welfare of the squabs was measured by behavioral measurement and good growth performance.

Materials and methods

Generally well written and the experimental design of how birds are included in different groups is easy to follow. Perhaps restructure and give subheadings. Explain and describe the sample population and its husbandry, housing (including sizes of cages) and nutrition first. Then explain the experimental set-up and what was tested on what (i.e. behavioural measures of welfare and growth /development parameters). Have a section with a subheading about the hand-rearing protocol. Then finished with a paragraph on statistical analysis.

Response:    the cage size was 1375px wide, 40 cm deep and 30 cm high for each unit, sectioning of material and method applied and subheading was added.

Were the foster pigeons that were used the same breed? And where they also in a reproductive phase? I.e. did they have eggs and nests? How did you get these foster pigeons to produce crop milk?

Response:  The foster parent was chosen of the same Egyptian Baladi breed, the foster pigeon was parent pigeon in the breeding cycle as it broods their own hatched squabs and we introduced the orphan hatched squabs to foster parent pigeon nest to be brooded by this foster parent. Foster parent was in breeding time so it secretes crop milk, but introducing hatched orphan squabs to foster parent nest increased the aggressive behavior of the parent toward those squabs or even neglecting them in feeding.    

I'm still unsure whether the human carers are really fostering? Perhaps you describe your groups as i) parent reared, ii) foster parent reared (pigeon), iii) hand-reared (human).

Response:  corrected.

Line 114: nesting material was provided in the form of...

Response:  corrected.

Line 121 and 122: please check your use of referencing as this hard to follow. Is this the style of the article? It would be easier to refer to a specific author so that you construct a proper sentence I.e. Pigeon parents were fed a standard pigeon ration as per Smith et al. [reference number].

Response:   the reference number was corrected, the specific nutrient requirement of pigeon is not tabulated in NRC books so I relay on researches as (Sales, J.; Janssens, G. Nutrition of the domestic pigeon (Columba livia domestica), World's poultry science journal 2003, 59, 221-232) to formulate balanced ration for adult pigeon.

Is table 1 needed? Surely this can be looked up from the reference about diets? Did you do any nutritional analysis that would warrant a breakdown of ingredients in this manner?

Response:   table 1 present the ration fed to pigeon during the experiment I relay on the reference (Sales, J.; Janssens, G. 2003) on nutrient requirement of pigeon mainly for crude protein and energy requirement and then I used the local feedstuff ingredient to formulate a ration to meet this requirement, then I calculate the metabolic energy, crude protein, and other ration total ingredients according to NRC 1994. 

Line 139: Vaseline is petroleum jelly. Is this safe to use on something an animal is ingesting?

Response:   we used medical Vaseline and it exhibits no side effect during the experiment, but in another study, we will replace it with more safe lubricant.

Line 145: how do pigeon squabs come of age?

Response:   corrected to grow up.

Line 149: please include your ethogram of behavioral definitions somewhere near here.

Response:   behavioral ethogram was included.

Line 151 and 152: I don't understand... each pigeon was watched for two hours? What's the three minutes for? If you do a focal sample, is this instantaneous or continuous recording? How often were behaviours noted? And did you focus on state or event behaviours? This needs more explanation. 

Response:   All the experimental groups were observed for a total of 2 hours per day. As each squab individually was observed for 3 minutes daily, the observation was continuous and we recorded the event behavior of the squab, not the state.

Line 153: according to whom? Is this a subheading? Please work on your layout.

Response:   corrected.

Line 154: this behavioural description does not make sense. I would not be able to observe this behaviour myself based on this definition and description.

Response:   more clarified.

Line 158: as above. Think about the name of this behaviour too.

Response:   clarified by head shaking.

Line 166-167: Re-write "As the squab increases with age (pin feathers developing) they defecate over the edge of the nest. When pin feathers are fully developed they move away from the nest to defecate"

Response:   corrected

Line 169: as above. I don't understand this behavior.

Response:   corrected to Bill snapping.

Line 172: How do you know this is why they do this?

Response:  Hissing and puffing: Squabs may fluff the feather especially the breast and ''hiss'' when accosted by an unfamiliar object or movement. They also flap their wings as a sign of flight stance to appear bigger in size and stand in biting position as a defense mechanism.

Line 174: again, how do you know this is why they do this?

Response:   Squeaker Notes: Squabs at 4 weeks of age or younger produce squeaky notes which adult pigeons term to be an alarm note when any strange thing close to their nest to inform their parents.

Line 185 and 186: strange referencing again / hard to read sentence structure.

Response:   corrected

Line 190: Only an ANOVA? No modelling was attempted to pick out the influences of different variables? Seems quite limited statistical analysis for an elaborate and complex experimental design.

Response:   the model added in statistical analysis.

Line 191: What do you mean? ref: no sig. diff.

Response:   The effect of replicate within the same group was analyzed and indicated no significant difference among replicates.

Duncan multiple range tests WERE used...

Response:   corrected

Results

You need to present statistical output correctly in the text (i.e. the test statistic, degrees of freedom, r2 value- if appropriate, and the P value). You do not provide the reader with any sense of the validity of your testing by just including P values.

Response:    p-value was corrected, degrees of freedom and F-values were added accordingly.

Think about whether or not you have really measured welfare... You measure behaviour and you collect information on growth. But you give no link to welfare. Are there any welfare-friendly behaviours that you would look out for that tell you pigeon squabs are experiencing positive welfare states?

Response:   poor welfare of the animals was measured by many parameters as low-performance parameters, high frequencies of stress behaviors and low survival rateThe link between our result and squabs welfare clarified in the discussion.

Please re-write the whole results section so that inferential analysis is presented around descriptive analysis for each hypothesis tested. 

Table 2: You are presenting multiple P values together for comparison and therefore you should check for Type I error, I suggest a Benjamini-Hochburg to correct your alpha level. See: Benjamini, Y. and Hochberg, Y. (1995) Controlling the False Discovery Rate: A practical and powerful approach to multiple testing. Journal of the Royal Statistical Society B 57 (1): 289–300. 

You should be able to find an automatic calculator for this online.

Response:   all result re-written and corrected to be easily followed.

Also, for table 2, what are these data being tested? Time? Frequency? Counts of behaviour? 

Response:   frequencies occurred during observation all over experimental time.

Table 3: I don't understand how these are growth indicators? They are behavioural categories again? Please clarify or change what data you include.

Response:   the order of tables was corrected.

Table 4: This is understandable. I see what is being analysed here. Again, work on statistical output.

Response:   statistics were repeated for table 4, 5.

Table 5: humans do not brood.

Response:   corrected.

Discussion

Please restructure. Start off with an explanation of your findings. Then tell the reader what they mean. And then add extensions. And finally your conclusion.

What are your key results and why? How do they compare to the background literature? How do they add to it?

At the moment, your discussion is hard to follow and does not "sell" the results of your own experiment to the reader.

Some of the discussion is very descriptive and simply explains pigeon natural history and physiology. You are not evaluating findings against published literature. Try to base your discussion on your results and synthesise new meaning from these results via comparison or contrast with published work. Or if your results are novel, and there is no comparison in the literature, evaluate why this is. Don't tell me about the physiology of how pigeons make crop milk, for example.

Response:   all discussion was rearranged and more clarified.

I am confused as to how foster pigeons were selected? Where they in breeding condition? In avicultural circles, pigeons make excellent foster parents so long as they are already nesting an they can be given eggs to hatch. Is the poor performance of foster parents because they were given already hatched chicks? Please explain this better. 

Response:   clarified in materials and method.

Line 267: "Squab behaviour stimulates..."

Response:   corrected.

Line 299: "This corresponds with data recorded by *Give author name*..."

Response:   corrected.

Line 300: improves survival

Response:   corrected.

Line 305: showed to be, not proved.

Response:   the survival rate was added and hand feeding of orphan squabs improved growth performance and survival rate than the foster pigeon.

Are the recommendations better embedded as a final paragraph of the discussion before the conclusion?

Response:    recommendations added to the discussion.

Line 314: hatched, not birthed.

Response:   corrected.

As I said previously, describe the repro. state of the foster pigeons. If they are in the right physiological state they will make excellent foster carers. 

We acknowledge the efforts of the reviewer and we corrected and edited the manuscript as recommended. All the comments contribute to improving the quality of our manuscript.

Round 2

Reviewer 1 Report

The revised manuscript is suitable for publication without significant changes. In particular, previous concerns  have been effectively addressed. I would only suggest some minor editing for language.

Author Response

Reviewer #1:

The revised manuscript is suitable for publication without significant changes. In particular, previous concerns have been effectively addressed. I would only suggest some minor editing for language.

Response: We appreciate the time, efforts, and overall constructive criticism raised by the reviewer. All the suggestions have been addressed, and corrections made where necessary. We revised the language and the whole writing.

Reviewer 2 Report

Much improved and more clarity added. A few basic grammatical points noted.

Just check your spelling and grammar all the way through. I have corrected some points below.

Line 49: "In Egypt, pigeons were reared in towers as domestic livestock". 

Line 55: "... as female pigeons reach..."

Line 57: "... leave the nest... The breeding cycle in pigeons..."

Line 59 Please check your format here, and put a space between begins consecutively. 

Line 162: Suggest correcting to: "Behavioral observations were performed using a focal sample technique [21], an observation sheet for recording behaviors based on an ethogram (see below), a stop watch and digital camera according to [22]. All the experimental groups were observed for a total of 2 hours per day. Each individual squabwas observed for 3 minutes daily and the observation was continuous to record event behaviors, detailed in the following."

Line 166: change parameters to Ethogram

Line 212: using a one-way ANOVA...

Line 253: Suggest "Regarding the survival rate for groups brooded by one parent (either male or female), this was significantly lower survival rate at 95% (degree of freedom 29 as each two squabs considered one replicate with 10 replicate for each 255 group) compared to that for the group brooded by both parents (100%).

Line 258; Suggest "These behaviors appear to be higher in groups brooded by one parent and this may be due to lower frequencies of feeding of these squabs. This lowered feeding rate may increasestress on squabs that leads to increased frequencies of these stress behaviors when compared to squab reared by two parents."

Line 265: Suggest "The hand-reared group recorded a significantly higher survival rate of 85%(degree of freedom 29 as each two squabs considered one replicate with 10 replicate for each group) compared to the foster pigeon reared group (65%) but a significantly lower survival rate than the two parent reared group (100%)."

Line 352: by fosters parents who may lack the motivation to adopt an unfamiliar squab.

Line 357: Hand-feeding by humans is shown to be...

Line 359: than foster parented squabs.

Author Response

Reviewer #2

Much improved and more clarity added. A few basic grammatical points noted.

Just check your spelling and grammar all the way through. I have corrected some points below.

Response:

We thank the reviewer for excellent reviewing and very useful suggestions to improve the quality of the manuscript. We acknowledge the efforts of the reviewer and we corrected and edited the manuscript as recommended. All the comments contribute to improving the quality of our manuscript.

Line 49: "In Egypt, pigeons were reared in towers as domestic livestock". 

Response: corrected.

Line 55: "... as female pigeons reach..."

Response: corrected.

Line 57: "... leave the nest... The breeding cycle in pigeons..."

Response: corrected.

Line 59 Please check your format here, and put a space between begins consecutively. 

Response: corrected.

Line 162: Suggest correcting to: "Behavioral observations were performed using a focal sample technique [21], an observation sheet for recording behaviors based on an ethogram (see below), a stop watch and digital camera according to [22]. All the experimental groups were observed for a total of 2 hours per day. Each individual squabwas observed for 3 minutes daily and the observation was continuous to record event behaviors, detailed in the following."

Response: corrected.

Line 166: change parameters to Ethogram

Response: corrected.

Line 212: using a one-way ANOVA...

Response: corrected.

Line 253: Suggest "Regarding the survival rate for groups brooded by one parent (either male or female), this was significantly lower survival rate at 95% (degree of freedom 29 as each two squabs considered one replicate with 10 replicate for each 255 group) compared to that for the group brooded by both parents (100%).

Response: corrected.

Line 258; Suggest "These behaviors appear to be higher in groups brooded by one parent and this may be due to lower frequencies of feeding of these squabs. This lowered feeding rate may increase stress on squabs that leads to increased frequencies of these stress behaviors when compared to squab reared by two parents."

Response: corrected, and moved to L284.

Line 265: Suggest "The hand-reared group recorded a significantly higher survival rate of 85%(degree of freedom 29 as each two squabs considered one replicate with 10 replicate for each group) compared to the foster pigeon reared group (65%) but a significantly lower survival rate than the two parent reared group (100%)."

Response: corrected.

Line 352: by fosters parents who may lack the motivation to adopt an unfamiliar squab.

Response: corrected.

Line 357: Hand-feeding by humans is shown to be...

Response: corrected.

Line 359: than foster parented squabs.

Response: corrected.

------

We repeat our appreciation to the Editor and the reviewers for their contribution in improving the quality of the manuscript and to meet the quality for publication in Animals.

This manuscript is a resubmission of an earlier submission. The following is a list of the peer review reports and author responses from that submission.